# Grammar Prompting for Domain-Specific Language Generation with Large Language Models

**Bailin Wang**[◇]  **Zi Wang**[†]  **Xuezhi Wang**[†]  **Yuan Cao**[‡]  **Rif A. Saurous**[†]  **Yoon Kim**[◇]

[◇]Massachusetts Institute of Technology  [†]Google DeepMind  [‡]Google Research

{bailinw, yoonkim}@mit.edu, {wangzi, xuezhiw, yuancao, rif}@google.com

## Abstract

Large language models (LLMs) can learn to perform a wide range of natural language tasks from just a handful of in-context examples. However, for generating strings from highly structured languages (e.g., semantic parsing to complex domain-specific languages), it is challenging for the LLM to generalize from just a few exemplars. We propose *grammar prompting*, a simple approach to enable LLMs to use external knowledge and domain-specific constraints, expressed through a grammar in Backus–Naur Form (BNF), during in-context learning. Grammar prompting augments each demonstration example with a specialized grammar that is minimally sufficient for generating the particular output example, where the specialized grammar is a subset of the full DSL grammar. For inference, the LLM first predicts a BNF grammar given a test input, and then generates the output according to the rules of the grammar. Experiments demonstrate that grammar prompting can enable LLMs to perform competitively on a diverse set of DSL generation tasks, including semantic parsing (SMCalFlow, Overnight, GeoQuery), PDDL planning, and SMILES-based molecule generation.

## 1 Introduction

Prompting large language models (LLMs) with demonstrations optionally combined with natural language instructions has been shown to be an effective approach for surfacing their myriad capabilities acquired through pretraining [10]. This approach is however inadequate for applications where the task specifications cannot be fully delineated through just a handful of exemplars, for example in semantic parsing where an LLM must translate a natural language utterance to an executable program in a domain-specific language (DSL). DSLs often incorporate domain-specific abstractions and semantics that are difficult to characterize via just a few demonstrations. And unlike general-purpose programming languages, DSLs are by definition specialized and thus unlikely to have been encountered often enough (or at all) during pretraining for the LLM to acquire its full syntax.

How can we draw on the few-shot learning capabilities of LLMs to generate structured strings that are substantially different from those seen during pretraining? This work explores *grammar prompting* as a simple approach for data-efficient generation of structured languages where an output string in the language can be derived through a series of symbolic manipulations. We exploit the fact that constraints over a structured output space can often be succinctly described by a context-free grammar in Backus–Naur Form (BNF), which is commonly used to define the syntax of a language. Grammar prompting augments each in-context example $(\boldsymbol{x}, \boldsymbol{y})$ with a *specialized* BNF grammar $G[\boldsymbol{y}]$ that is minimally sufficient for generating $\boldsymbol{y}$. Given a new input, the LLM first predicts the specialized BNF grammar and then generates the answer conditioned on the grammar.

Grammar prompting follows the recent line of work which enhances the few-shot reasoning capabilities of LLMs by interleaving intermediate "reasoning" steps between each in-context input and

---

Code and data available at: https://github.com/berlino/grammar-prompting.

37th Conference on Neural Information Processing Systems (NeurIPS 2023).

output [51, 24, 86, 80, 73]. The key difference in our approach is that the intermediate variable is in the form of a formal grammar rather than in natural language, which focuses on eliciting the symbolic manipulation capabilities of LLMs. The use of a formal grammar moreover makes it possible to impose constraints during incremental decoding such that syntactic validity is guaranteed. Finally, unlike chain-of-thought-style prompts [86] which typically require manual verbalization of the intermediate reasoning steps, in our approach the specialized grammar $G[\boldsymbol{y}]$ can be derived automatically by parsing the output $\boldsymbol{y}$ with the full (unspecialized) DSL grammar.

To summarize,

- We propose grammar prompting as a simple approach for enabling LLMs to generate highly structured languages from just a few exemplars.

- We design a constrained LLM decoding algorithm tailored to grammar prompting, which guarantees syntactic validity while minimizing the number of LLM API calls.

- We apply grammar prompting to various domain specific languages for semantic parsing (SM-CalFlow, Overnight, GeoQuery), AI planning (PDDL), and molecule generation (SMILES), and find that it can meaningfully improve upon standard prompting baselines in the few-shot setting.

## 2 Background

In this section, we define our problem and review the few-shot learning method that we build on.

### 2.1 Problem Formulation: Domain-Specific Language Generation

Let $\Sigma^*$ be the set of all finite strings over an alphabet $\Sigma$, and further let $D \subseteq \Sigma^*$ be a domain-specific language (DSL) for an application of interest. Given an input $\boldsymbol{x}$ (e.g., a natural language command) we are interested in generating $\boldsymbol{y} \in D$ (e.g., a program in a DSL fulfilling the command), as shown by the following calendar assistant example from SMCalFlow [6]:

$\boldsymbol{x}$ : Add meeting with Jean's manager on Wednesday at 3PM.

$\boldsymbol{y}$ : `CreateEvent(& (start_? WednesdayNumberPM(3))(attendee_? FindManager(Jean)))`

DSLs are crafted by experts who use their domain-specific knowledge to incorporate higher-level abstractions than are typically found in general-purpose programming languages. We assume access to an expert-defined grammar $G$ that fully specifies the DSL's syntax. As is the case with many DSLs, we further assume that $G$ is a context-free grammar in Backus–Naur Form (BNF). See Figure 1 for a simple example adapted from SMCalFlow [6]. Letting $L(G)$ be the language generated by $G$, we have $D \subseteq L(G) \subseteq \Sigma^*$ (not all syntactically valid programs are semantically valid).

```
event       ::=   "CreateEvent(" constraint ")"
            |     "QueryEvent(" constraint ")"
constraint  ::=   "(&" constraint constraint ")"
            |     "(start_?" date time? ")"
            |     "(attendee_?" attendee* ")"
date        ::=   "Wednesday" | "Monday"
number      ::=   ("0".."9")+
time        ::=   "NumberAM(" number ")"
            |     "NumberPM(" number ")"
attendee    ::=   | "Bob" | "Carol" | "Jean"
            |     "FindManager"
```

**Figure 1:** A simple BNF grammar for a calendar DSL.

### 2.2 Few-shot Learning with Large Language Models

In-context learning with large language models (LLMs) has been shown to be an effective approach for few-shot learning [10]. Under this approach, a pretrained LLM is conditioned on $N$ demonstration examples $(\boldsymbol{x}^{(i)}, \boldsymbol{y}^{(i)})_{i=1}^N$ followed by a test example $\boldsymbol{x}$, and the output is given by decoding from the prompted LLM, i.e., $P_{\text{LLM}}(\boldsymbol{y} \mid \boldsymbol{x}, (\boldsymbol{x}^{(i)}, \boldsymbol{y}^{(i)})_{i=1}^N)$. The demonstration examples can be optionally preceded by natural language instructions to further improve performance or even enable zero-shot learning [85, 62]. Recent work has additionally shown that interleaving natural language verbalizations of intermediate reasoning steps between each $\boldsymbol{x}^{(i)}$ and $\boldsymbol{y}^{(i)}$ can greatly improve few-shot performance on complex reasoning tasks [51, 86, 80, 73, 16].

The effectiveness of few-shot in-context learning depends both on how useful the implicit knowledge acquired through pretraining is for the task, and on how effectively the task specifications can be conveyed through the demonstrations. For DSL, the structured nature of combinatorial output space (i.e., the DSL grammar $G$) cannot be adequately captured through just a handful of demonstrations. Thus, few-shot generation of strings of a DSL remains challenging for LLMs.

```
LLM Prompt

You are an expert programmer, and you need to write a program for the given natural language query.
First, you should write a grammar that contains all the necessary BNF rules. Then, you should write
programs that conform to your predicted rules.
----------------------------------------------------------------------------------------------------
(optional) G:        [BEGIN RULES]                    . . .                    [END RULES]
----------------------------------------------------------------------------------------------------
x⁽¹⁾: find the meeting on Wednesday with Bob and Carol
----------------------------------------------------------------------------------------------------
G[y⁽¹⁾]:            event         ::=    "QueryEvent(" constraint ")"
                    constraint    ::=    "(&" constraint constraint ")"
                                   |     "(start_?" date ")"
                                   |     "(attendee_?" attendee attendee ")"
                    date          ::=    "Wednesday"
                    attendee      ::=    "Bob" | "Carol"
----------------------------------------------------------------------------------------------------
y⁽¹⁾: QueryEvent(& (start_? Wednesday)(attendee_? Bob Carol))
----------------------------------------------------------------------------------------------------
                                              . . .
----------------------------------------------------------------------------------------------------
x: Add meeting with Jean's manager on Wednesday at 3PM
```

```
LLM Output

Ĝ:                  event         ::=    "CreateEvent(" constraint ")"
                    constraint    ::=    "(&" constraint constraint ")"
                                   |     "(start_?" date time ")"
                                   |     "(attendee_?" attendee ")"
                    date          ::=    "Wednesday"
                    time          ::=    "NumberPM(3)"
                    attendee      ::=    "FindManager(" attendee ")" | "Jean"
----------------------------------------------------------------------------------------------------
ŷ: CreateEvent(& (start_? Wednesday NumberPM(3))(attendee_? FindManager(Jean)))
```

**LLM Prompt**

You are an expert programmer, and you need to write a program for the given natural language query. First, you should write a grammar that contains all the necessary BNF rules. Then, you should write programs that conform to your predicted rules.

(optional) $G$:        [BEGIN RULES]                    . . .                    [END RULES]

$x^{(1)}$: find the meeting on Wednesday with Bob and Carol

$G[y^{(1)}]$:
```
event         ::=    "QueryEvent(" constraint ")"
constraint    ::=    "(&" constraint constraint ")"
               |     "(start_?" date ")"
               |     "(attendee_?" attendee attendee ")"
date          ::=    "Wednesday"
attendee      ::=    "Bob" | "Carol"
```

$y^{(1)}$: QueryEvent(& (start_? Wednesday)(attendee_? Bob Carol))

. . .

$x$: Add meeting with Jean's manager on Wednesday at 3PM

**LLM Output**

$\widehat{G}$:
```
event         ::=    "CreateEvent(" constraint ")"
constraint    ::=    "(&" constraint constraint ")"
               |     "(start_?" date time ")"
               |     "(attendee_?" attendee ")"
date          ::=    "Wednesday"
time          ::=    "NumberPM(3)"
attendee      ::=    "FindManager(" attendee ")" | "Jean"
```

$\widehat{y}$: CreateEvent(& (start_? Wednesday NumberPM(3))(attendee_? FindManager(Jean)))

**Figure 2:** Example of grammar prompting for a calendar DSL. We interleave the minimal specialized grammar $G[y^{(i)}]$ between the demonstrations $x^{(i)}$ and $y^{(i)}$. During decoding, the LLM first predicts the specialized grammar $\widehat{G}$, and then predicts the program $\widehat{y}$ conditioned on $\widehat{G}$. The blue portion is not part of the actual prompt and only shown for illustrative purposes.

## 3 Grammar Prompting

Grammar prompting exploits the fact that while the actual strings of a DSL may not have been encountered frequently enough (or at all) during pretraining for the LLM to implicitly acquire its syntax, the LLM will likely have encountered many instances of *metalanguages* (languages used to describe other languages). BNF grammars are a standard metalanguage for specifying a language's syntax, and are expected to occur in the LLM training corpus with some frequency (e.g., in computer science textbooks). We thus focus on using BNF grammars for few-shot DSL generation.

Let $G = \bigcup_{j=1}^{M} \{r_j\}$ be an extended BNF grammar where each rule $r_j$ is of the form

```
<symbol> ::= <expr₁> | <expr₂> | ...
```

Here `<symbol>` is a nonterminal symbol and each `<expr₁>` is a sequence of nonterminal and terminal symbols.[1] A straightforward approach for incorporating a BNF grammar during in-context learning is to simply prepend the string representation of the full grammar $G$ to the demonstration examples, along with an instruction to use the grammar. However in preliminary experiments, we found that this did not yield any improvements.[2]

### 3.1 Specialized Grammars

We propose to use *specialized grammars* to enable better use of domain-specific knowledge and constraints. A specialized grammar $G' \subseteq G$ is a grammar obtained from taking a subset of the rules of the full grammar $G$. We further define $G[y]$, a *minimal specialized grammar* of $y$, to be a BNF

---

[1]For brevity we forgo the formal tuple-based definition of $G$ and instead define $G$ to be equivalent to its context-free rules. We also freely go back and forth between this set definition of $G$ and its string representation.

[2]However, when combined with specialized grammars we did observe small improvements by appending the full DSL grammar to the instructions. Hence, for all experiments where $G$ is small enough (GeoQuery, Overnight-B, SMILES), we include $G$ as part of the instruction. See Figure 2.

grammar with the following properties: (1) $\boldsymbol{y} \in L(G[\boldsymbol{y}])$, and (2) $\forall r \in G[\boldsymbol{y}], \ \boldsymbol{y} \notin L(G[\boldsymbol{y}] \setminus \{r\})$.[3] We can readily obtain a minimal specialized grammar by using $G$ to parse $\boldsymbol{y}$ and then taking the union of rules that were used in the derivation of $\boldsymbol{y}$.

Grammar prompting feeds a sequence of $(\boldsymbol{x}^{(i)}, G[\boldsymbol{y}^{(i)}], \boldsymbol{y}^{(i)})_{i=1}^{N}$ along with $\boldsymbol{x}$ as a prompt to an LLM. For inference we first obtain the specialized grammar with an (approximate) $\arg\max$ decoding

$$\widehat{G} = \underset{G' \subseteq G}{\arg\max} \ P_{\mathrm{LLM}}(G' \mid \boldsymbol{x}, (\boldsymbol{x}^{(i)}, G[\boldsymbol{y}^{(i)}], \boldsymbol{y}^{(i)})_{i=1}^{N}).$$

We then obtain the program conditioned on $\widehat{G}$,

$$\widehat{\boldsymbol{y}} = \underset{\boldsymbol{y} \in L(\widehat{G})}{\arg\max} \ P_{\mathrm{LLM}}(\boldsymbol{y} \mid \widehat{G}, \boldsymbol{x}, (\boldsymbol{x}^{(i)}, G[\boldsymbol{y}^{(i)}], \boldsymbol{y}^{(i)})_{i=1}^{N}).$$

We discuss how to perform constrained decoding with $\widehat{G} \subseteq G$ and $\widehat{\boldsymbol{y}} \in L(\widehat{G})$ in the next section. Grammar prompting views DSL program generation as a *grammar specialization* process where given a natural language specification $\boldsymbol{x}$, a set of production rules, $\widehat{G}$, is selected from $G$, and then a program $\widehat{\boldsymbol{y}}$ is deduced according to the selected rules. Grammar prompting can also be viewed as an instance of chain-of-thought prompting [51, 86] where the intermediate thought is in the form of a formal grammar. However, unlike typical chain-of-thought prompting where the answer is (usually) deterministic given the intermediate reasoning steps, in our case there is still some uncertainty with respect to $\widehat{\boldsymbol{y}}$ given $\widehat{G}$ (e.g., $L(\widehat{G})$ could still be infinite).

### 3.2 Constrained Decoding

The use of a formal grammar as an intermediate variable makes it possible to enforce grammatical constraints during autoregressive LLM decoding. We first discuss how we enforce the constraint $\boldsymbol{y} \in L(\widehat{G})$. One approach to constrained decoding is to use $\widehat{G}$ to obtain a left-to-right Earley parser [18] and only decode from valid continuations at each decoding step. However this simple strategy poses several practical challenges when working with API-only LLMs. For one, a valid terminal continuation in $\widehat{G}$ may consist of multiple BPE tokens. Moreover, while we can sample a valid continuation at each time step by disallowing invalid tokens,[4] since the set of valid continuations changes at each time step, this strategy would require calling the LLM API at each time step with the full

---

**Algorithm 1** Earley-based Constrained Generation

---

**Input**: Test input $\boldsymbol{x}$, predicted grammar $\widehat{G}$
**Output**: Program $\hat{y} \in L(\widehat{G})$

1: $\hat{y} \leftarrow \epsilon$      ▷ initialize to empty string
2: **while** True **do**
3:    $\bar{y} \leftarrow \mathrm{decode}\left(P_{\mathrm{LLM}}(\cdot \mid \boldsymbol{x}, \widehat{G}, \hat{y}, \dots)\right)$
4:    $\hat{y} \leftarrow \hat{y} \cdot \bar{y}$      ▷ concatenation
5:    **if** $\hat{y} \in L(\widehat{G})$ **then**    ▷ try parsing with $\widehat{G}$
6:      **return** $\hat{y}$      ▷ return if successful
7:    **else**    ▷ if parsing fails, need to correct
8:      $\boldsymbol{y}_{\mathrm{prefix}}, \Sigma[\boldsymbol{y}_{\mathrm{prefix}}] \leftarrow \mathrm{EarleyParse}(\hat{y}, \widehat{G})$
9:      $\boldsymbol{w}^* \leftarrow \underset{\boldsymbol{w} \in \Sigma[\boldsymbol{y}_{\mathrm{prefix}}]}{\arg\max} P_{\mathrm{LLM}}(\boldsymbol{w} \mid \boldsymbol{y}_{\mathrm{prefix}}, \dots)$
10:      $\hat{y} \leftarrow \boldsymbol{y}_{\mathrm{prefix}} \cdot \boldsymbol{w}^*$
11:    **end if**
12: **end while**
13: **return** $\hat{y}$

---

prompt and prefix along with the disallowed continuations, which is prohibitively expensive.[5]

While there are many methods for grammar-constrained LM decoding [68, 64, 26], we present a simple strategy which speculatively decodes from the LLM to look ahead for multiple tokens. The pseudocode is shown in Algorithm 1. At each prediction step, we ask the LLM to speculatively decode the full program conditioned on the current prefix (lines 4-5). If the resulting continuation leads to a valid program, we return it (lines 6-7). Otherwise, we consult an Earley parser to extract the longest valid prefix from the current prediction ($\boldsymbol{y}_{\mathrm{prefix}}$), along with a set of valid terminals that can

---

[3]Note that $\boldsymbol{y}$ may have more than one minimal specialized grammar due to the potential instantiation of extended BNF rules. For instance, the expression `"(attendee_?" attendee+ ")"` depicted in Figure 1 implicitly defines all occurrences of attendee greater than 1. If this expression is incorporated into a program, either the concrete rule `"(attendee_?" attendee ")"` or the original rule could be included in the minimal specialized grammar. In most applications we consider the rules of the minimal specialized grammar will be concrete, and thus there will only be one parse associated with $\boldsymbol{y}$. See appendix A.1 for further details.

[4]For example by using the `logit_bias` argument from OpenAI's LLM API.

[5]These costs might be mitigated in the future if LLM APIs allow for cheaper use of cached prompts.



**Earley-based Constrained Decoding**

$\hat{\boldsymbol{y}}^{(t)}$ : CreateEvent(& (start_? Wednesday NumberPM(3))(attendee_? *Jean's* Manager))

----

*Extract the longest valid prefix, and possible fixes (i.e., next terminals) based on Earley parsing*:

$\boldsymbol{y}_{\text{prefix}}$ : QueryEvent(& (start_? Wednesday)(attendee_?

$\Sigma[\boldsymbol{y}_{\text{prefix}}]$: {*Jean* , *FindManager(*}

----

*Find the best candidate and concatenate it with the prefix*:

$\hat{\boldsymbol{y}}_{(t)} \leftarrow$ QueryEvent(& (start_? Wednesday)(attendee_? FindManager(



**Figure 3:** Illustration of how an predicted program is corrected in our proposed Earley-based constrained decoding. The final partial program will be subsequently fed into the LLM for continuation.

follow the prefix ($\Sigma[\boldsymbol{y}_{\text{prefix}}]$). Finally, we rely on the LLM's probabilities to decide which terminal to use, with which a new partial program can be constructed (lines 10-11).[6] Figure 3 illustrates one prediction step where the predicted program is corrected into a new valid partial program. Note that $\boldsymbol{w}$ can consist of multiple BPE tokens, e.g., "FindManager(" in Figure 3. By scoring over multi-token terminals, the search procedure is implicitly augmented by looking ahead for a few tokens.

We use a similar procedure to operationalize the constraint $G' \subseteq G$, except that EarleyParse (used at Algorithm 1, line 9) is constructed with a *metagrammar* (i.e., the grammar of $G$) for grammar prediction. See appendix A.1 for more details. In our ablation study we find that while these constraints are helpful insofar as they guarantee syntactic validity, grammar prompting still meaningfully improves upon standard prompting with even with simple unconstrained decoding.

## 4 Experiments

We apply grammar prompting to diverse domains: DSLs for semantic parsing (SMCalFlow, Overnight, GeoQuery), an action DSL (PDDL planning), and a molecule generation DSL (SMILES). These experiments are not necessarily intended to improve upon the state-of-the-art on these benchmarks but rather intended to assess whether LLMs can improve upon standard prompting for few-shot DSL generation by learning to predict and use grammars during in-context learning.

### 4.1 Semantic Parsing for Tool Usage

Software tools are typically accompanied by a collection of human-interpretable APIs which provide a platform for developers to interact programmatically with the tools. These APIs constitute a DSL, where each production rule of the grammar specifies the input and output types for a specific API call (see Figure 1 for an example). These tools demonstrate a broad spectrum in terms of DSL complexity, ranging from single-function tools such as Google(user_query), Translate(sentence, language) to more complex tools such as the entirety of Wolfram language.[7] Enabling LLMs to use external tools via APIs is an important step towards enhancing their capabilities [63, 56, 53, 72, 47].

We test our approach on standard semantic parsing benchmarks involving complex DSLs: SM-CalFlow [6], which features human-generated utterances about calendar management (see Figure 2); GeoQuery [99] which features queries against a US Geography database; and Overnight-Blocks [81], which features queries about blocks in a synthetic block world. See appendix B for examples of input-output pairs along with the specialized grammars. The original benchmarks target the training of conventional semantic parsers and thus contain hundreds/thousands of training examples. Even prompting-based approaches on these benchmark rely on retrieval-based in-context learning which first retrieves $m$ exemplars from a large training set of $n$ examples ($n \gg m$) based on some similarity measure (e.g., BM-25), and then performs in-context learning with the retrieved exemplars [57, 95, 68, 46]. In contrast, we target the true few-shot setting where we only assume access to 16–32 demonstration examples.

Our baselines here include: (1) standard prompting, (2) standard prompting with constrained decoding based on the full DSL grammar $G$ [68, 64], and (3) a derivation tree-based prompting baseline which imbues more structural information to the exemplars by feeding the linearized derivation

----

[6]In rare cases the set $\Sigma[\boldsymbol{y}_{\text{prefix}}]$ was too large to feed to LLM APIs. In these cases we used Sentence-BERT [59] to compute the similarity between the encoding of $\boldsymbol{y}_{\text{prefix}} \cdot \boldsymbol{w}$ and $\hat{\boldsymbol{y}}^{(t)}$ and took the top 16 candidates as $\Sigma[\boldsymbol{y}_{\text{prefix}}]$.

[7]https://www.wolfram.com/language/

| Approach | GeoQuery Prog. | GeoQuery Exec. | SMCalFlow Prog. | Overnight-Blk Prog. | Overnight-Blk Exec. |
|---|---|---|---|---|---|
| Standard Prompting (unconstrained decoding) | 60.7 | 81.5 | 46.4 | 29.3 | 54.7 |
| *w. constrained decoding ($\widehat{y} \in L(G)$)* | 61.1 | 81.8 | 49.2 | 29.3 | 54.7 |
| Linearized Derivation Tree Prompting | 58.6 | 77.5 | 50.0 | 27.3 | 56.4 |
| Grammar Prompting (unconstrained decoding) | 67.1 | 87.5 | 50.8 | 34.8 | 57.4 |
| *w. grammar constraint ($\widehat{G} \subseteq G$)* | 67.9 | 88.6 | 51.3 | 37.1 | 60.4 |
| *w. grammar and program constraint ($\widehat{y} \in L(\widehat{G})$)* | 69.6 | 88.9 | 52.4 | 37.6 | 60.9 |
| *w. oracle grammar ($\widehat{G} = G[y]$)* | 95.7 | 96.1 | 80.0 | 73.9 | 94.2 |
| *w. oracle grammar + program constraint* | 95.7 | 96.8 | 83.6 | 74.4 | 96.5 |

**Table 1:** Results on few-shot semantic parsing with Codex with various decoding strategies. GeoQuery and Overnight-Blk use 32 in-context examples, and SMCalFlow uses 16 examples. We show both program (Prog.) and execution (Exec.) accuracy when possible.

| Model (# ICL examples / # retrieval set) | Retrieval-based ICL | | | GeoQuery Out-of-Distribution | | | |
|---|---|---|---|---|---|---|---|
| | GeoQuery (32/560) | SMCalFlow (16/128) | Overnight-Blk (32/1,436) | Template (32/441) | TMCD (32/440) | Length (32/440) | NewFunc (32/453) |
| Previous Work | 86.1♣ | 60.7♠ | 65.2◇ | – | – | – | – |
| Standard Prompting | 96.8 | 60.0 | 69.4 | 93.2 | 77.1 | 86.4 | 63.3 |
| Grammar Prompting | 97.9 | 62.8 | 70.2 | 95.7 | 86.6 | 88.6 | 90.8 |
| *w. oracle grammar* | 98.6 | 88.9 | 97.2 | 97.9 | 95.0 | 95.7 | 96.2 |

**Table 2:** Results on retrieval-based in-context learning (left) and compositional generalization (right) with Codex. GeoQuery and Overnight-Blk show execution accuracy while SMCalFlow shows program accuracy. The numbers with ♣, ♠ and ◇ are taken from Herzig and Berant [31], Ye et al. [95] and Cao et al. [11], respectively.

tree instead of the surface form program.[8] We use Codex-davinci-002 [13] as the base LLM for these main experiments. Language models trained on code (such as Codex) have shown to be particularly effective on semantic parsing benchmarks [67]. We evaluate according to program accuracy (matching the predicted and reference programs) as well as execution accuracy (same execution in both programs) if possible.

**Few-shot results.** The main results are shown in Table 1. We find that grammar prompting can meaningfully improve upon the standard prompting baseline even without constrained decoding. Interestingly, grammar prompting outperforms derivation tree prompting which actually provides *more* information than the minimal specialized grammar $G[y]$ (since the derivation tree explicitly shows how the rules are actually applied to obtain the program). This potentially indicates that having the LLM "plan out" the program by forcing it to predict the specialized grammar $\widehat{G}$ first is an effective strategy. We also analyze the effect of constrained decoding on the number of LLM API calls in Table 7 of appendix A.1, where we observe that constrained decoding requires roughly three times more API calls than unconstrained decoding. However, despite the promising performance of grammar prompting, there is a large gap between using the predicted grammar versus using the oracle grammar (i.e., setting $\widehat{G} = G[y]$), indicating opportunities for further work in this area.

**Retrieval-based in-context learning.** While our core target application is few-shot semantic parsing, we also apply grammar prompting for retrieval-based in-context learning to test whether it can still improve performance in the data-rich regime and also to compare against prior work on these benchmarks. Results in Table 2 (left) show that grammar prompting can improve results even in this setting, although the improvements are less pronounced than in the few-shot setting.

**Out-of-distribution generalization.** We experiment to see whether grammar prompting can improve compositional generalization on GeoQuery. Specifically, we test grammar prompting on the compositional splits of GeoQuery split from Shaw et al. [66]. These splits feature structural divergence between training and test examples, e.g., programs have different templates or length. Results in Table 2 (right) shows that grammar prompting can improve upon standard prompting, across all splits (Template, TMCD, Length).

---

[8]For example, the derivation tree of a subprogram (attendee_? FindManager(Jean)) is linearized to [constraint "(attendee_?" [attendee "FindManager(" [attendee "Jean" ")"] ")"], which uses square brackets to encode richer hierarchical information than just the surface form program.

We next assess whether grammar prompting can enable LLMs to make zero-shot use of *unseen functions* (NewFunc) that are not even part of the retrieval set. We set aside 8 functions (`smallest`, `shortest`, `most`, `highest`, `sum`, `population_1`, `count`, `major`) and remove them from the retrieval set, simulating a scenario where new functions are supported in the backend yet no NL-program paired data is available for adapting a semantic parser. Note that for GeoQuery (and Overnight-Blk), we always prepend the full DSL grammar $G$—which includes the held-out functions—before the in-context exemplars. Table 2 (right-most column) shows that grammar-prompted LLMs achieve significantly better performance than standard prompting. Our results suggest that the explicit prediction of specialized grammars elicits understanding and reasoning at the grammar level, thereby enabling generalization to unseen functions. We also found that without constrained generation, LLMs were often able to guess functions that did not exist but were nonetheless sensible. An interesting direction is to explore whether LLMs can tackle DSL-open benchmarks such as LARC [1].

**Different base LLMs.** We finally experiment with grammar prompting across different base LLMs. Since GPT-3.5's 4K token limit is smaller than Codex's (8K) and GPT-4's (8K) limits, we use fewer exemplars in these experiments than before (24/8/16 exemplars for GeoQuery/SMCalFlow/Overnight-B respectively). Due to API cost, we limit our experiments to a smaller subset of 100 test examples instead of the full test set.

Table 3 shows that grammar prompting improves upon standard prompting in the majority of the settings. The exceptions are

| Base LM | Method | GeoQuery | SMCalFlow | Overnight-Blk |
|---|---|---|---|---|
| Codex | Standard | 83 | 27 | 63 |
| | Grammar | 95 | 35 | 66 |
| GPT-3.5 | Standard | 75 | 9 | 49 |
| | Grammar | 86 | 5 | 67 |
| GPT-4 | Standard | 85 | 32 | 56 |
| | Grammar | 98 | 36 | 62 |
| PaLM 2-L | Standard | 90 | 14 | 59 |
| | Grammar | 87 | 17 | 62 |

**Table 3:** Results with different base LLMs on a subset of 100 examples sampled from the original test set. GeoQuery and Overnight-Blk show execution accuracy, while SMCalFlow shows program accuracy.

SMCalFlow with GPT-3.5 where both methods performed poorly, and GeoQuery with PaLM 2-L[7], where standard prompting already performed well.

## 4.2 Class-Specific Molecule Generation

We next demonstrate an application of grammar prompting beyond language parsing problems with a molecule generation task. Existing methods for molecule generation typically focus on training specialized neural models using large training sets [45, 37, 15, 2, 79, 61, 19]. We instead follow Guo et al. [29] and explore a few-shot setting where the task is to generate class-specific molecules given a small number of exemplars of that class. Formally, given a small set of molecules $\{\boldsymbol{y}_c^{(i)}\}_{i=1}^N$ belonging to a particular molecule class $c \in \{\texttt{Acrylates}, \texttt{Chain Extenders}, \texttt{Isocyanates}\}$, our goal is to generate novel molecules $\boldsymbol{y}_c$ of the same class that can be synthesized using existing molecules. Since the in-context examples in this case will only consist of molecules of the same class, the "input" $\boldsymbol{x}_c^{(i)}$ is the empty string in this case. The data contains 32 Acrylates, 11 Chain Extenders, and 11 Isocyanates (see appendix G of Guo et al. [29]).

While molecules can be more faithfully represented with 3D graph structure, the SMILES string representation [87] remains popular due to its ease of use.[9] The specialized grammars $G[\boldsymbol{y}_c]$ (which are specialized from the SMILES grammar) encode various structural properties of the molecule that are specific to the molecule class. Figure 4 shows an example of a specialized grammar and the corresponding molecule in SMILES format. In this example, `ring_closure ::= "1"` specifies the number of rings, and `branch ::= "(" smiles ")"` specifies whether there is a branch.

We test our approach by generating 100 molecules for each class and assessing the quality of the generated molecules. In addition to the standard prompting baseline, we also run the graph grammar baseline from Guo et al. [29] which learns a hypergraph grammar [38] from the given molecules. We use four metrics: *Validity (V)*, the percentage of chemically valid molecules; *Diversity (D)*, average pairwise Tanimoto distance over Morgan fingerprints [60]; *Retrosynthesis score (R)*, the percentage

---

[9]Note that SMILES does not guarantee that a generated string corresponds to a valid molecule. Using our approach on more advanced string representations such as SELFIES [44] (which guarantee validity) remains an interesting avenue for future work.

```
Specialized SMILES Grammar for Molecule Generation
```

$G[\boldsymbol{y}]$:
```
        smiles          ::=    atom chain branch chain chain | atom chain
        atom            ::=    organic_symbol
        organic_symbol  ::=    "C" | "N" | "O"
        chain           ::=    atom ring_closure bond atom | bond atom
                        |      bond atom ring_closure | atom
                        |      atom bond atom bond atom
                        |      bond atom bond atom
        ring_closure    ::=    "1"
        bond            ::=    "="
        branch          ::=    "(" smiles ")"
```

$\boldsymbol{y}$: CC(= C)C(= O)OCCOC1 = CC = CC = C1

**Figure 4:** Example of a specialized grammar for generating a molecule from the Acrylates class.

| Model | Acrylates | | | | Chain Extenders | | | | Isocyanates | | | |
|---|---|---|---|---|---|---|---|---|---|---|---|---|
| | V | D | R | M | V | D | R | M | V | D | R | M |
| Graph Grammar [29] | 100 | 0.83 | 79.0 | 30.3 | 100 | 0.86 | 72.7 | 98.3 | 100 | 0.93 | 52.2 | 82.7 |
| Standard Prompting | 87.7 | 0.73 | 80.0 | 76.7 | 60.3 | 0.89 | 72.7 | 55.7 | 94.7 | 0.82 | 78.0 | 92.2 |
| Grammar Prompting | 98.0 | 0.74 | 91.0 | 93.3 | 96.3 | 0.90 | 86.7 | 94.0 | 97.7 | 0.79 | 78.0 | 96.3 |

**Table 4:** Results for few-shot molecule generation with GPT-3.5. The metrics are validity (V), diversity (D), retrosynthesis score (R) and membership (M). Higher is better for all metrics.

of molecules that are synthesizable from existing molecules, which is computed approximately via the Retro* model [12]; *Membership (M)*, the percentage of molecules that belong to the desired monomer class. We use GPT-3.5 as the base LLM and sample from the LLM without constrained decoding, as constrained decoding was found to decrease the diversity of samples. See appendix A.2 for the full experimental setup.

**Results.** Table 4 shows that, compared with standard prompting, grammar prompting significantly improves the synthesis of Acrylates and Chain Extenders across all metrics, while yielding mixed results for Isocyanates. Notably, both prompting-based methods outperform the graph grammar baseline in terms of Retro score, possibly due to that LLMs may have been pre-trained on existing datasets of molecules, enabling them to effectively generate synthesizable molecules. In contrast, the baseline method cannot incorporate any external knowledge beyond the 11 or 32 molecules provided. Our preliminary results indicate that LLMs can serve as a useful tool for generating string representations of chemical structures (and potentially other biological/chemical structures).

### 4.3 Action Subset Selection for Efficient PDDL Planning

Our final experiments show how grammar prompting can improve the efficiency of classical AI planners. Classical planning is the problem of finding a sequence of actions (i.e., a plan) that goes from an initial state $\boldsymbol{s}_0$ to a goal state $\boldsymbol{s}_g$. An action is represented by a ground operator (e.g., unstack(block1, block2) which consists of an operator unstack along with two object arguments). We additionally consider *macro-operators* which can potentially speed up planning [9].[10] Planning tasks, along with actions, are represented in Planning Domain Definition Language (PDDL) [27]. We explore how grammar prompted LLMs can help guide classical planning algorithms.

We design specialized grammars to provide guidance to the classical greedy best-first search (GBFS) algorithm [5] by selecting a set of relevant actions. Figure 5 illustrates an example of such a specialized grammar, which captures all the necessary actions for the final plan $\boldsymbol{y}$ that solves the given task. The process of the guided planning consists of the following steps: (1) given a task, predict a specialized grammar $G[\boldsymbol{y}]$; (2) use the specialized grammar $G[\boldsymbol{y}]$ to subsequently generate a plan within the restricted action space derived from $G[\boldsymbol{y}]$; (3) initialize GBFS's priority queue with the LLM-generated plan, and (4) search for the final plan in the restricted action space. Our setup builds upon the idea of using an LLM-generated plan to initialize GBFS from Silver et al. [69], which has a simpler two-step process: (1) given a task, predict a plan via standard prompting, and (2) utilize this plan to guide GBFS. We use their method as our baseline.

Following Silver et al. [69], we create a similar few-shot setting for LLM planning, using 5 tasks as in-context examples and 10 tasks for evaluation from Pyperplan [5]. We test our approach on 3 classic

---

[10]For example, pickup-and-stack(A, B) is a combination of pickup(A) and stack(A, B).

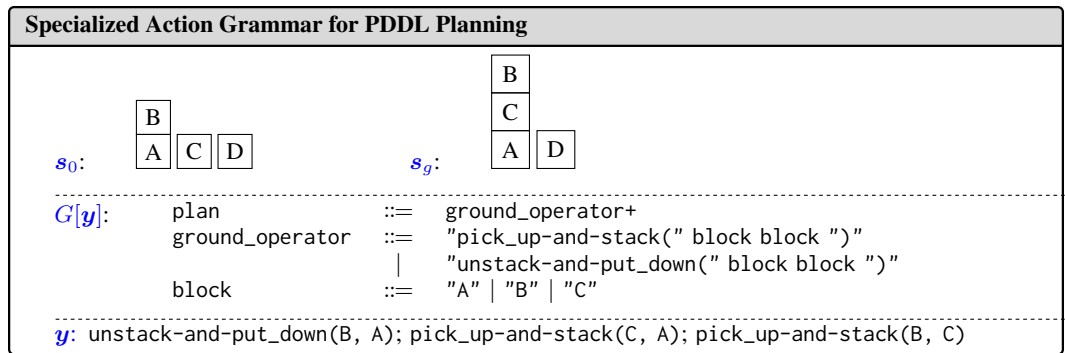

**Figure 5:** Example of a specialized grammar for PDDL planning in the Blocks domain. Given an input $x = (s_0, s_g)$, the specialized grammar $G[y]$ only includes necessary actions for solving this task.

| Approach | Blocks | | | Depot | | | Satellite | | |
|---|---|---|---|---|---|---|---|---|---|
| | Created | Expanded | Success | Created | Expanded | Success | Created | Expanded | Success |
| GBFS + Prim. (No LLM) | 360 | 188 | 1.0 | 18047 | 3870 | 0.4 | 8205 | 150 | 1.0 |
| Standard + Prim. | 348 | 180 | 1.0 | 17597 | 4039 | 0.4 | 6686 | 78 | 1.0 |
| Grammar + Prim. | 251 | 124 | 1.0 | 15033 | 3641 | 0.4 | 5162 | 64 | 1.0 |
| Standard + Macro. | 850 | 16 | 1.0 | 1460 | 56 | 0.4 | 4003 | 27 | 0.3 |
| Grammar + Macro. | 170 | 9 | 1.0 | 2917 | 127 | 0.8 | 3665 | 46 | 0.9 |
| Standard + Min Macro. | 228 | 8 | 1.0 | 1903 | 65 | 0.6 | 3483 | 35 | 0.8 |

**Table 5:** Results on PDDL planning. Created/Expanded refer to the number of nodes during planning (lower is better). Success refers to success rate (higher is better). Numbers are averaged over three runs using GPT-3.5.

domains in PDDL planning, including Blocks, Depot and Satellite. For the action space, we use either a set of primitive actions (Prim) or an augmented set with macro actions (Macro). In addition to standard prompting, we add two more baselines: (1) No LLM: planning with the entire set of actions; (2) Min Macro: where we construct a minimal set of macro actions for each domain by selecting actions from existing plans for the training tasks. The Min Macro baseline is a domain-specific method to reduce the action space. By comparing to Min Macro, we can verify the effectiveness of instance-specific v.s. domain-specific action selection. See appendix A.3 for more details.

**Results.** We evaluate the efficiency of planning in terms of the number of search nodes created/-expanded, as well as the success rate. Table 5 shows the promising performance of LLM-guided planning via grammar prompting. In Blocks, grammar prompting significantly improves efficiency while maintaining 100% success rate. In Depot, grammar prompting with macro actions improves the success rate by 20% over the best competing baseline. In Satellite, using primitive actions yields the best performance with 100% success rate and a reduction of 57% expanded nodes comparing to the No LLM baseline. While our experiments are not intended to complete with the state-of-the-art algorithms for fast planning [20–22, 32, 25, 84], they indicate the promise of LLMs for improving existing planning algorithms.

## 5 Discussion and Limitations

We discuss several limitations of our approach including some negative results. Grammar prompting did not yield any improvements for DSLs that were likely to have been frequently encountered during pretraining (e.g., regular expressions, SQL). Moreover, constrained generation based on specialized grammars led to increased API calls, and was not always beneficial for tasks beyond semantic parsing. For instance, in molecule generation we discovered that enforcing constraints can sometimes result in lower diversity. Additionally, in PDDL planning we observed that the constraints applied to prune objects can sometimes negatively impact performance, suggesting that relevant object selection is still very challenging for LLMs. It may be interesting to explore whether finetuning of moderately-sized LLMs using specialized grammars can lead to better grammar-based models for DSL generation.

On the positive front, our work demonstrates that LLMs have the capacity to understand and generate metalanguages. Working in this "metalanguage space" can be combined with chain-of-thought-style [86] prompts by, for example, manually providing natural language comments to the rules of the specialized grammars. We found this to improve results slightly on semantic parsing (see Figure 6

of appendix A.1). Moreover, many scientific problems can be formally approached by representing hypotheses as DSL programs [71], and DSLs can enable easier encoding of human prior knowledge and scientific principles, providing a foundation for scientific discovery. Recent work shows that state-of-the-art LLMs can follow previously unseen formal systems [75]. Techniques like grammar prompting can widen the scope of scientific problems for which LLMs could be effectively applied by more explicitly accounting for external knowledge and constraints.

## 6 Related Work

**Chain-of-thought prompting.** Grammar prompting extends a recent line of work on improving reasoning capabilities by requesting explicit reasoning steps as part of the prompt [51, 24, 86, 80, 14, 94]. Our approach is closely related to concurrent work on employing symbolic variables as part of the prompt [30, 50, 33, 97, 52], though we are not aware of any existing work that uses formal grammars as the intermediate reasoning step.

**LLMs for program generation and semantic parsing.** Generating programs from natural language specifications, a task often referred to as semantic parsing, is a sub-problem of program synthesis; for surveys, see Kamath and Das [39] and Gulwani et al. [28]. Recent works [8, 89] have explored using LLMs for generating code in general-purpose programming languages (e.g., Python). Our work further extends this line by examining whether LLMs can generate DSL programs, which are intrinsically scarce. There has also been work on using LLMs for tool usage via further training [63] or prompting [56, 77], investigating how model scales [57] and retrievers [96, 46] affect in-context learning for semantic parsing, and constrained decoding [64, 68, 55] for program generation.

**Neural grammars.** Grammar prompting can also been seen as a "fully LLM" instantiation of a line of work on neural parameterizations of symbolic grammars [35, 17, 43, 42, 36, 100, 92, 91, 93]. Indeed, our approach to semantic parsing essentially uses prompt-based learning to define a quasi-synchronous grammar [70, 78] whose rules dynamically depend on the source sentence. Concretely, in contrast to recent works which embed learnable neural components within synchronous grammars [41, 23, 76], grammar prompting relies on the implicit in-context learning capabilities of LLMs for the learning component. (However unlike these works, our conditional grammar does not explicitly align its rules to the subparts of the source sentence).

**Grammar-based molecule generation.** Grammar-based methods have gained significant interest in the realm of molecule generation, offering advantages in interpretability, data-efficiency, and controllability. One line of research involves integrating generic SMILES grammars with neural networks to generate syntactically correct molecules [45, 15]. Another approach centers on data-driven induction of grammars for generation [29, 38]. Our work aligns with the former, viewing grammar prompting as a straightforward method for integrating grammar into an LLM without the need for additional training.

**LLMs for planning.** Recently, LLMs have been increasingly studied in the context of planning for autonomous agents. When given goals expressed in natural language in household environments, earlier works [3, 65, 34, 48] directly prompted LLMs to predict executable actions. However, in PDDL domains, recent works [69, 74] showed that LLMs underperform classical planners if the desired action sequences are very long. Grammar prompting represents a promising strategy for augmenting classical planners with LLMs to get the best of both worlds. Other related efforts include translating between problems and PDDL models [49] and corrective re-prompting [58]. Besides using LLMs, integrating learning and planning has been extensively studied in the past literature, e.g., learning actions [4, 82], skills [84], macro-actions [9], rules [88] and guidance strategies [90, 83, 40] for more efficient planning.

## 7 Conclusion

We propose grammar prompting as a simple approach for improving few-shot DSL generation with large language models. Experiments across a range of structured languages including DSLs for semantic parsing (SMCalFlow, GeoQuery, Overnight), PDDL planning (action DSL), and molecule generation (SMILES), show that grammar prompting can improve upon standard prompting baselines. The encouraging results in semantic parsing indicate its potential to assist LLMs with tool usage, and the promising results in other domains indicate that grammar prompting can enable application of LLMs in domains that intrinsically depend on DSLs.

## Acknowledgments

We thank Jacob Andreas, Gabriel Grand, Linlu Qiu, Tom Silver, and Hunter Lang for helpful discussion and feedback. This study was supported by funds from the Google-MIT research collaborations program and the GIST-MIT joint research program.

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

| | **Few-Shot** | | | **Retrieval-based** | | | **GeoQuery Out-of-Dist.** | | | |
| | GeoQuery | SMCalflow | Overnight-Blk | GeoQuery | SMCalflow | Overnight-Blk | Template | TMCD | Length | NewFunc |
|---|---|---|---|---|---|---|---|---|---|---|
| Train | 32 | 16 | 32 | 560 | 128 | 1436 | 441 | 440 | 440 | 453 |
| Test | 280 | 360 | 399 | 280 | 360 | 399 | 439 | 440 | 440 | 447 |

**Table 6:** Statistics of the splits used for experiments on semantic parsing.

| Approach | **GeoQuery** | **SMCalFlow** | **Overnight-B** |
|---|---|---|---|
| Standard Prompting (unconstrained decoding) | 81.5 (1.0) | 46.4 (1.0) | 54.7 (1.0) |
| *w. constrained decoding ($\widehat{\boldsymbol{y}} \in L(G)$)* | 81.8 (4.3) | 49.2 (5.6) | 54.7 (1.6) |
| Linearized Derivation Tree Prompting | 77.5 (1.0) | 50.0 (1.0) | 56.4 (1.0) |
| Grammar Prompting (unconstrained decoding) | 87.5 (1.0) | 50.8 (1.0) | 57.4 (1.0) |
| *w. grammar constraint ($\widehat{G} \subseteq G$)* | 88.6 (3.0) | 51.3 (3.0) | 60.4 (1.4) |
| *w. grammar and program constraint ($\widehat{\boldsymbol{y}} \in L(\widehat{G})$)* | 88.9 (3.3) | 52.4 (3.3) | 60.9 (2.8) |
| *w. oracle grammar ($\widehat{G} = G[\boldsymbol{y}]$)* | 96.1 (1.3) | 80.0 (1.0) | 94.2 (1.0) |
| *w. oracle grammar + program constraint* | 96.8 (2.1) | 83.6 (2.6) | 96.5 (1.0) |

**Table 7:** Extended results which show the number of Codex calls per example on few-shot semantic parsing in brackets. Columns in grey show program accuracy, while white columns others indicate execution accuracy.

# A  Experiment Details

## A.1  Semantic Parsing

**Statistics and Splits.**  We show the statistics for the splits used for the experiments in Table 6.

For GeoQuery, we utilize the standard split from Zelle and Mooney [99] in the retrieval-based setting and the Template, TMCD, and Length splits from Shaw et al. [66]. We randomly sample 32 examples from the training set of the standard split to create the few-shot split. To generate the NewFunc split, we designate examples utilizing the following eight functions as test examples: smallest, shortest, most, highest, sum, population_1, count, major; the remaining examples are incorporated into the training set.

For SMCalFlow, we adopt the 16-shot and 128-shot cross-domain settings from Yin et al. [98] as our few-shot and retrieval-based settings, respectively. It is noteworthy that the training set of the original splits contains approximately 25k in-domain training examples. Previous work [96, 57] utilized these examples as their retrieval set. However, in our experiments, we found that incorporating in-domain examples did not enhance performance. Consequently, we use 16/128 cross-domain examples as our training set in the few-shot and retrieval settings, respectively. For all experiments on SMCalFlow, we used the preprocessed version from Qiu et al. [57], which employs a more concise LISPRESS format [54] than the original version [98]. This format aligns with Ye et al. [96] for a fair comparison.

For Overnight-Blocks, we employ the standard split from Wang et al. [81] in the retrieval setting. We randomly sample 32 examples from the training set of the standard split to create the few-shot split.

**Scoring Functions for Constrained Generation.**  For each candidate continuation $\boldsymbol{w} \in \Sigma[\boldsymbol{y}_{\text{prefix}}]$ for correction, we first form a partial program via concatenation $\boldsymbol{y}_{\text{prefix}} \cdot \boldsymbol{w}$ and then feed it into Codex to obtain the score for the candidate via

$$\log P_{\text{LLM}}(\boldsymbol{w} \mid \widehat{G}, \boldsymbol{x}, \boldsymbol{y}_{\text{prefix}}, (\boldsymbol{x}^{(i)}, G[\boldsymbol{y}^{(i)}], \boldsymbol{y}^{(i)})_{i=1}^{N}).$$

In the case where $\boldsymbol{w}$ consists of multiple BPE tokens, e.g., FindManger( is tokenized into Find, Manager, and (, we average the token-level log-likelihood to obtain a candidate-level score. However, when the size of $\Sigma[\boldsymbol{y}_{\text{prefix}}]$ exceeds 16, invoking Codex for each candidate becomes too expensive. To address this issue, we employ SentenceBERT to select 16 most plausible candidates first via a dot product,

$$(\text{SentenceBERT}(\hat{\boldsymbol{y}}_t))^{\top} (\text{SentenceBERT}(\boldsymbol{y}_{\text{prefix}} \cdot \boldsymbol{w})),$$

where SentenceBERT yields the embeddings for the incorrect prediction $\hat{\boldsymbol{y}}_t$ and a candidate of corrected partial program $\boldsymbol{y}_{\text{prefix}} \cdot \boldsymbol{w}$.

| Molecule Class | Temperature | Presence Penalty | Frequency Penalty |
|---|---|---|---|
| *Sampling specialized grammars $\widehat{G}$* | | | |
| Acrylates | 0.6 | 0.1 | 0.1 |
| Chain Extenders | 0.6 | 0.1 | 0.1 |
| Isocyanates | 0.6 | 0.4 | 0.4 |
| *Sampling molecules $\widehat{y}$* | | | |
| Acrylates | 0.6 | 0.1 | 0.1 |
| Chain Extenders | 0.6 | 0.1 | 0.1 |
| Isocyanates | 0.3 | 0.4 | 0.4 |

**Table 8:** Hyperparameters for sampling specialized grammars $\widehat{G}$ (top) and the molecules $\widehat{y}$ in grammar prompting for molecule generation. Standard prompting uses the same hyperparameters for $y$.

The functionality of obtaining the log-likelihood for a candidate continuation given a prefix is applicable via Codex APIs [11] via setting `logprobs=True` and `echo=True`. Unfortunately, subsequent models (e.g., GPT-3.5 and GPT-4) disable such functionality. As a workaround, we simply use the scoring function based on SentenceBERT to directly select the best candidate in our PDDL planning experiments.

**Cost Efficiency.** We assess various decoding strategies for their cost efficiency, focusing on the number of API calls. The number of Codex calls resulting from the few-shot semantic parsing experiments is presented in Figure 7, alongside the corresponding accuracy metrics. The results indicate that standard prompting under constrained decoding leads to a significantly higher number of Codex calls. Similarly, grammar-prompting with constraints also results in an increased number of Codex calls. However, when employing both grammar and program constraints, the number of calls decreases meaningfully in comparison to standard prompting under constrained decoding. Future work might consider exploring strategies for more cost-efficient constrained decoding.

**Grammar Prompting with Annotated Rules.** We have additionally experimented with enhancing BNF rules by appending natural language comments. As illustrated in Figure 6, we pair each BNF rule with its corresponding natural language phrases extracted from the given query $x$. These manually annotated comments yield an explicit correspondence between natural language phrases and their corresponding BNF rules, thereby better facilitating interpretation and application of the grammars for generating programs. When employing the augmented grammar prompting, we noticed marginal improvements on SMCalFlow (+1.0%) and Overnight-Blocks (0.5%), with no observed enhancement on GeoQuery. While the gains might not appear significant, this predicted alignment could potentially contribute to improved interpretability and further constraints on generation. For instance, the phrase "someone's manager" should consistently trigger the function `FindManager(`. We leave the exploration of utilizing the augmented rules for future work.

## A.2 Molecule Generation

**Sampling Procedure** Different from semantic parsing and PDDL planning, where the most probable program $y$ needs to be found via $\arg\max$ inference, molecule generation has empty specification $x$ and requires sampling from a prompting-based distribution. The sampling procedure for grammar prompting consists of three stages: (1) we randomly sample a permutation of given molecules, denoted as $\pi$, (2) based on a prompt formed by the permutation, we sample a specialized grammar $\widehat{G}$ via

$$\widehat{G} \sim P_{\text{LLM}}(G' \mid x, (G[y^{(\pi[i])}], y^{(\pi[i])})_{i=1}^{N}),$$

iii) we finally obtain the molecule conditioned $\widehat{G}$,

$$\widehat{y} \sim P_{\text{LLM}}(y \mid \widehat{G}, (G[y^{(\pi[i])}], y^{(\pi[i])})_{i=1}^{N}).$$

We list the hyperparameters used for the sampling procedure in for (2) in Table 8 (top) and for (3) in Table 8 (bottom).

---

[11] https://learn.microsoft.com/en-us/azure/cognitive-services/openai/reference

```
LLM Prompt

You are an expert programmer, and you need to write a program for the given natural language query.
First, you should write a grammar that contains all the necessary BNF rules. Then, you should write
programs that conform to your predicted rules.
```

$x^{(1)}$: find the meeting on Wednesday with Bob and Carol

$\check{G}[y^{(1)}]$:
```
        event       ::=    "QueryEvent(" constraint ")"              find the meeting
        constraint  ::=    "(&" constraint constraint ")"
                      |    "(start_?" date ")"                        on ...
                      |    "(attendee_?" attendee attendee ")"       with ...
        date        ::=    "Wednesday"                                Wednesday
        attendee    ::=    "Bob" | "Carol"                            Bob and Carol
```

$y^{(1)}$: QueryEvent(& (start_? Wednesday)(attendee_? Bob Carol))

$\cdot\cdot\cdot$

$x$: Add meeting with Jean's manager on Wednesday at 3PM

```
LLM Output
```

$\widehat{G}$:
```
        event       ::=    "CreateEvent(" constraint ")"             Add meeting
        constraint  ::=    "(&" constraint constraint ")"
                      |    "(start_?" date time ")"                   on ... at ...
                      |    "(attendee_?" attendee ")"                 with ...
        date        ::=    "Wednesday"                                Wednesday
        time        ::=    "NumberPM(3)"                              3PM
        attendee    ::=    "FindManager(" attendee ")" | "Jean"      Jean's manager
```

$\widehat{y}$: CreateEvent(& (start_? Wednesday NumberPM(3))(attendee_? FindManager(Jean)))

**Figure 6:** Example of grammar prompting where BNF grammars are additionally annotated with natural
language comments (shown in green). These manually curated comments provide a detailed mapping between
natural language phrases and their corresponding BNF rules, thereby better facilitating interpretation and
application of the grammars for generating programs. We manually craft and add these comments to the few-shot
prompts (top). The model predicts this during inference (bottom).

In comparison, the sampling procedure for standard prompting only consists of two stages: (1) we
randomly sample a permutation of given molecules, denoted as $\pi$, (2) based on a prompt formed by
the permutation, we directly sample a molecule via

$$\widehat{y} \sim P_{\text{LLM}}(y \,|\, (y^{(\pi[i])})_{i=1}^{N}).$$

The hyperparameters used for Step (2) is the same as in grammar prompting and shown in Table 8
(bottom).

While we observed that Earley-based constrained generation enhances grammar prompting in terms
of improving validity, other metrics, such as the retrosynthesis score, decreased significantly. This
discrepancy could be attributed to the fact that existing LLMs, due to their limited exposure to
molecules represented in SMILES format, struggle with comprehending and applying the BNF
grammar rules of SMILES. Overall, our current findings serve as preliminary evidence that grammar
prompting can tap into the capacity of LLMs to understand and apply BNF rules. However such
capacity still remains limited in text-focused LLMs.

### A.3 PDDL Planning

**Restricted Action Space** The specialized grammar defined for PDDL planning essentially delin-
eates a constrained action space that includes necessary actions and their associated objects. Our
empirical results found that limiting the classical GBFS planner to the objects selected by a spe-
cialized grammar proved too restrictive, yielding beneficial results only within the Blocks domain.
Therefore, we decided to remove this limitation, thus expanding the action space of GBFS to contain
the actions predicted from the grammar with an unrestricted range of objects.

# B  Prompts

Figures 7, 8, and 9 demonstrate the prompts with grammars, based on actual examples in the SMCalFlow, GeoQuery, and Overnight datasets respectively. Because the general grammar of SMCalFlow is long (around 4k tokens), we do not include it within the prompt. For GeoQuery and Overnight, the general grammar is integrated as part of the instruction within the prompt. In the context of molecule generation, the general grammar for SMILES[12] is also included. Figures 10 and 11 demonstrate the prompts with action DSLs for PDDL planning.

---

[12]https://metamolecular.com/cheminformatics/smiles/formal-grammar/

---

**LLM Prompt**

You are an expert programmer, and you need to write a program for the given natural language query. First, you should write a grammar that contains all the necessary BNF rules. Then, you should write programs that conform to your predicted rules.

query: I need a meeting with Elli and her team on Wednesday afternoon .

BNF grammar rules:

```
call                 ::=   "(Yield" event ")"
event                ::=   "(CreatePreflightEventWrapper" event_constraint ")"
event_constraint     ::=   "(&" event_constraint event_constraint ")"
                      |    "(Event.start_?" datetime_constraint ")"
                      |    "(Event.attendees_?" attendee_constraint ")"
datetime_constraint  ::=   "(DateTimeConstraint" time date ")"
time                 ::=   "(Afternoon)"
date                 ::=   day
day                  ::=   "(NextDOW" day ")" | DAY
DAY                  ::=   "(Wednesday)"
attendee_constraint  ::=   "(&" attendee_constraint attendee_constraint ")"
                      |    "(AttendeeListHasRecipient" person ")"
                      |    "(AttendeeListHasPeople" person ")"
person               ::=   recipient | "(FindTeamOf" person ")"
recipient            ::=   "(Execute (refer (extensionConstraint" recipient ")))"
                      |    "(RecipientWithNameLike (PersonName.apply" string "))"
string               ::=   ""Elli""
```

program based on the BNF grammar rules:
(Yield(CreatePreflightEventWrapper(&(Event.start_?(DateTimeConstraint(Afternoon) (NextDOW(Wednesday)))))(Event.attendees_?(&(AttendeeListHasRecipient(Execute(refer (extensionConstraint(RecipientWithNameLike(PersonName.apply "Elli")))))) (AttendeeListHasPeople(FindTeamOf(Execute(refer(extensionConstraint (RecipientWithNameLike(PersonName.apply "Elli")))))))))))

. . .

query: Create a meeting tomorrow with the staff .

---

**LLM Output**

BNF grammar rules:
```
call                 ::=   "(Yield" event ")"
event                ::=   "(CreatePreflightEventWrapper" event_constraint ")"
event_constraint     ::=   "(&" event_constraint event_constraint ")"
                      |    "(Event.start_?" datetime_constraint ")"
                      |    "(Event.attendees_?" attendee_constraint ")"
datetime_constraint  ::=   "(DateTime.date_?" datetime_constraint ")"
                      |    "(?" OP datetime ")"
OP                   ::=   "="
datetime             ::=   date
date                 ::=   day
day                  ::=   DAY
DAY                  ::=   "(Tomorrow)"
attendee_constraint  ::=   "(AttendeeListHasPeople" person ")"
person               ::=   "(FindReports" person ")"
                      |    recipient
recipient            ::=   "(toRecipient (CurrentUser))"
```

program based on the BNF grammar rules:
(Yield(CreateCommitEventWrapper(CreatePreflightEventWrapper(&(Event.start_? (DateTime.date_?(? = (Tomorrow))))(Event.attendees_?(AttendeeListHasPeople (FindReports(toRecipient(CurrentUser)))))))))

**Figure 7:** Prompt with real examples from the SMCalFlow dataset.

**LLM Prompt**

You are an expert programmer, and you need to write a program for the given natural language query. First, you should write grammar rules by choosing from the following BNF rules. Then, you should write programs that conform to your predicted rules.

-----------------------------------------------------------------------------------------------

[BEGIN RULES]

```
query        ::=   "answer(" answer_type ")"
answer_type  ::=   city | state | num | place | river | country
city         ::=   "city(" city ")"
                 | "cityid('" CITYNAME "', '" STATEABBREV "')"
                 | "cityid('" CITYNAME "', _)"
                 | "capital(" city ")"
                 | "major(" city ")"
                 | "capital_1(" state ")"
                 | "loc_2(" state ")"
                 | "loc_2(" country ")"
                 | "largest(" city ")"
                 | "smallest(" city ")"
                 | "intersection(" city "," city ")"
                 | "exclude(" city "," city ")"
                 | "largest_one(population_1(" city "))"
                 | "largest_one(density_1(" city "))"
                 | "smallest_one(population_1(" city "))"
                 | "smallest_one(density_1(" city "))"
                 | ALL_CITY

                           ...
```

 [END RULES]

-----------------------------------------------------------------------------------------------

query: what states border hawaii ?
BNF grammar rules:

```
query        ::=   "answer(" answer_type ")"
answer_type  ::=   state
state        ::=   "state(" state ")"
                 | "next_to_2(" state ")"
                 | "stateid('" STATENAME "')"
STATENAME    ::=   "hawaii"
```

-----------------------------------------------------------------------------------------------

program based on the BNF grammar rules:
`answer(state(next_to_2(stateid('hawaii'))))`

-----------------------------------------------------------------------------------------------
                                       ...
-----------------------------------------------------------------------------------------------

query: how many major cities are in arizona ?

---

**LLM Output**

BNF grammar rules:

```
query        ::=   "answer(" answer_type ")"
answer_type  ::=   num
num          ::=   "count(" city ")"
city         ::=   "major(" city ")"
                 | "city(" city ")"
                 | "loc_2(" state ")"
state        ::=   "stateid('" STATENAME "')"
STATENAME    ::=   "arizona"
```

-----------------------------------------------------------------------------------------------

program based on the BNF grammar rules:
`answer(count(major(city(loc_2(stateid('arizona'))))))`

**Figure 8:** Prompt with real examples from the GeoQuery dataset.

**LLM Prompt**

You are an expert programmer, and you need to write a program for the given natural language query. First, you should write grammar rules by choosing from the following BNF rules. Then, you should write programs that conform to your predicted rules.

------------------------------------------------------------------------------------------------------

[BEGIN RULES]

```
    list_value         ::=    "(listValue" list_value ")"
                          |    "(filter" list_value PROPERTY ")"
                          |    "(filter" list_value PROPERTY OP list_value ")"
                          |    "(superlative" list_value AGGREGATE "
                          |    (ensureNumericProperty" PROPERTY "))"
                          |    "(countSuperlative" list_value AGGREGATE PROPERTY ")"
                          |    "(_size" list_value ")"
                          |    "(aggregate" AGGREGATE list_value ")"
                          |    "(getProperty" list_value PROPERTY ")"
                          |    "(getProperty (singleton" SINGLETON_VALUE ") !type)"
                          |    "(concat" ENTITY_VALUE ENTITY_VALUE ")"
                          |    "(concat" NUMBER_VALUE NUMBER_VALUE ")"
                          |    ENTITY_VALUE | NUMBER_VALUE
    PROPERTY           ::=    "shape" | "color" | "length" | "is_special" | "width"
                          |    "height" | "left" | "right" | "above" | "below"
                          |    "(reverse left)" | "(reverse right)"
                          |    "(reverse above)" | "(reverse below)"
    SINGLETON_VALUE    ::=    "en.block" | "en.shape" | "en.color"
    ENTITY_VALUE       ::=    "en.block.block1" | "en.block.block2" | "en.shape.pyramid"
                          |    "en.shape.cube" | "en.color.red" | "en.color.green"
    NUMBER_VALUE       ::=    "3 en.inch" | "6 en.inch" | "2"
    OP                 ::=    "=" | ">" | "<" | ">=" | "<=" | "!="
    AGGREGATE          ::=    "sum" | "max" | "min" | "avg"
```

[END RULES]

------------------------------------------------------------------------------------------------------

query: which brick is taller than brick 1
BNF grammar rules:

```
    list_value         ::=    "(listValue" list_value ")"
                          |    "(aggregate" AGGREGATE list_value ")"
                          |    "(getProperty" list_value PROPERTY ")"
                          |    "(getProperty (singleton" SINGLETON_VALUE ") !type)"
    AGGREGATE          ::=    "avg"
    SINGLETON_VALUE    ::=    "en.block"
    PROPERTY           ::=    "width"
```

------------------------------------------------------------------------------------------------------

program based on the BNF grammar rules:
(listValue(aggregate avg (getProperty(getProperty(singleton en.block) !type) width)))

------------------------------------------------------------ · · · ------------------------------------------------------------

query: look right of block 1 and get the special block

---

**LLM Output**

BNF grammar rules:

```
    list_value         ::=    "(listValue" list_value ")"
                          |    "(filter" list_value PROPERTY OP list_value ")"
                          |    "(filter" list_value PROPERTY ")"
                          |    "(getProperty (singleton" SINGLETON_VALUE ") !type)"
                          |    ENTITY_VALUE
    SINGLETON_VALUE    ::=    "en.block"
    PROPERTY           ::=    "is_special"
                          |    "(reverse left)"
    OP                 ::=    "="
    ENTITY_VALUE       ::=    "en.block.block1"
```

------------------------------------------------------------------------------------------------------

program based on the BNF grammar rules:
(listValue(filter(filter(getProperty(singleton en.block) !type) is_special)
    (reverse left) = en.block.block1))

**Figure 9:** Prompt with real examples from the Overnight dataset. Some long rules are omitted to fit the space.

**LLM Prompt**

**Q:**
(:objects a b c d e - block)
(:init (clear d) (clear c) (ontable d) (ontable a) (on c e) (on e b) (on b a) (handempty))
(:goal (on a e) (on e b) (on b d) (on d c))
**DSL:**
```
                    plan     ::=    action+
                    action   ::=    "(unstack" object object ")"
                             |      "(put-down" object ")"
                             |      "(pick-up-and-stack" object object ")"
                             |      "(unstack-and-stack" object object object ")"
                    object   ::=    "c" | "e" | "d" | "b" | "a"
```
**A:**
(unstack c e) (put-down c) (pick-up-and-stack d c) (unstack e b) (put-down e) (unstack-and-stack b a d) (pick-up-and-stack e b) (pick-up-and-stack a e)

------------------------------------------------------------------------------------------------------------------

**Q:**
(:objects a b c d - block)
(:init (clear a) (clear c) (clear d) (ontable a) (ontable b) (ontable d) (on c b) (handempty))
(:goal (on a b) (on b c) (on c d))
**DSL:**
```
                    plan     ::=    action+
                    action   ::=    "(unstack-and-stack" object object object ")"
                             |      "(pick-up-and-stack" object object ")"
                    object   ::=    "c" | "b" | "d" | "a"
```
**A:**
(unstack-and-stack c b d) (pick-up-and-stack b c) (pick-up-and-stack a b)
------------------------------------------------------------------------------------------------------------------
                                                · · ·
------------------------------------------------------------------------------------------------------------------

**Q:**
(:objects a b c d - block)
(:init (clear c) (clear a) (clear b) (clear d) (ontable c) (ontable a) (ontable b) (ontable d) (handempty))
(:goal (on d c) (on c b) (on b a))
**DSL:**

**LLM Output**
```
                    plan     ::=    action+
                    action   ::=    "(pick-up-and-stack" object object ")"
                    object   ::=    "b" | "a" | "c" | "d"
```
**A:**
(pick-up-and-stack b a) (pick-up-and-stack c b) (pick-up-and-stack d c)

**Figure 10:** Prompt with real examples in the Blocks domain from Pyperplan. The prompt template follows [69].

---

**LLM Prompt**

**Q:**
(:objects crate0 crate1 crate2 crate3 crate4 crate5 depot0 distributor0 distributor1 hoist0 hoist1 hoist2 pallet0 pallet1 pallet2 pallet3 pallet4 pallet5 truck0 truck1 - object)
(:init (pallet pallet0) (surface pallet0) (at pallet0 depot0) (clear crate5) (pallet pallet1) (surface pallet1) (at pallet1 distributor0) (clear pallet1) (pallet pallet2) (surface pallet2) (at pallet2 distributor1) (clear crate3) (pallet pallet3) (surface pallet3) (at pallet3 distributor0) (clear pallet3) (pallet pallet4) (surface pallet4) (at pallet4 distributor0) (clear crate4) (pallet pallet5) (surface pallet5) (at pallet5 distributor1) (clear crate1) (truck truck0) (at truck0 distributor1) (truck truck1) (at truck1 depot0) (hoist hoist0) (at hoist0 depot0) (available hoist0) (hoist hoist1) (at hoist1 distributor0) (available hoist1) (hoist hoist2) (at hoist2 distributor1) (available hoist2) (crate crate0) (surface crate0) (at crate0 distributor0) (on crate0 pallet4) (crate crate1) (surface crate1) (at crate1 distributor1) (on crate1 pallet5) (crate crate2) (surface crate2) (at crate2 distributor1) (on crate2 pallet2) (crate crate3) (surface crate3) (at crate3 distributor1) (on crate3 crate2) (crate crate4) (surface crate4) (at crate4 distributor0) (on crate4 crate0) (crate crate5) (surface crate5) (at crate5 depot0) (on crate5 pallet0) (place depot0) (place distributor0) (place distributor1))
(:goal (on crate0 pallet3) (on crate1 crate4) (on crate3 pallet1) (on crate4 pallet5) (on crate5 crate1))
**DSL:**

```
plan      ::=   action+
action    ::=   "(lift-and-drive" object object object object object object ")"
          |     "(drive-and-load" object object object object ")"
          |     "(drive-and-lift" object object object object object ")"
          |     "(lift" object object object object ")"
          |     "(drive-and-unload" object object object object object ")"
          |     "(drop" object object object object ")"
object    ::=   "truck1" | "hoist0" | "crate5" | "pallet0" | "depot0" | "truck0" | "hoist2"
          |     "crate4" | "crate0" | "distributor1" | "hoist1" | "pallet4" | "distributor0"
          |     "crate1" | "pallet5" | "crate3" | "crate2" | "pallet1" | "pallet3"
```

**A:**
(lift-and-drive truck1 hoist0 crate5 pallet0 depot0 depot0) (drive-and-load truck1 hoist0 crate5 depot0) (drive-and-lift truck0 hoist2 crate4 crate0 distributor1) (lift hoist1 crate0 pallet4 distributor0) (drive-and-lift truck1 hoist0 crate1 pallet5 depot0) (drive-and-load truck1 hoist0 crate1 depot0) (drive-and-lift truck1 hoist0 crate3 crate2 depot0) (drive-and-load truck1 hoist0 crate3 depot0) (drive-and-unload truck1 hoist0 crate3 pallet1 depot0) (drop hoist2 crate4 pallet5 distributor1) (drive-and-unload truck1 hoist0 crate1 crate2 depot0) (drive-and-lift truck0 hoist2 crate1 crate2 distributor1) (drop hoist2 crate1 crate4 distributor1) (drive-and-unload truck1 hoist0 crate5 crate1 depot0) (drop hoist1 crate0 pallet3 distributor0)

----------------------------------------------------------------------------------------------------
                                                  . . .
----------------------------------------------------------------------------------------------------

**Q:**
(:objects crate0 crate1 crate2 crate3 depot0 distributor0 distributor1 hoist0 hoist1 hoist2 pallet0 pallet1 pallet2 truck0 truck1 - object) (:init (pallet pallet0) (surface pallet0) (at pallet0 depot0) (clear crate0) (pallet pallet1) (surface pallet1) (at pallet1 distributor0) (clear crate3) (pallet pallet2) (surface pallet2) (at pallet2 distributor1) (clear crate2) (truck truck0) (at truck0 depot0) (truck truck1) (at truck1 depot0) (hoist hoist0) (at hoist0 depot0) (available hoist0) (hoist hoist1) (at hoist1 distributor0) (available hoist1) (hoist hoist2) (at hoist2 distributor1) (available hoist2) (crate crate0) (surface crate0) (at crate0 depot0) (crate crate1) (surface crate1) (at crate1 distributor1) (on crate1 pallet2) (crate crate2) (surface crate2) (at crate2 distributor1) (on crate2 crate1) (crate crate3) (surface crate3) (at crate3 distributor0) (on crate3 pallet1) (place depot0) (place distributor0) (place distributor1)) (:goal (on crate0 pallet2) (on crate1 crate3) (on crate2 pallet0) (on crate3 pallet1))
**DSL:**

---

**LLM Output**

```
plan      ::=   action+
action    ::=   "(lift-and-drive" object object object object object object ")"
          |     "(lift" object object object object ")"
          |     "(drive" object object object ")"
          |     "(drive-and-lift" object object object object object ")"
          |     "(drop" object object object object ")"
          |     "(drive-and-load" object object object object ")"
          |     "(drive-and-unload" object object object object object ")"
object    ::=   "truck0" | "hoist0" | "crate0" | "pallet0" | "depot0" | "hoist2"
          |     "crate2" | "crate1" | "distributor1" | "distributor0" | "hoist1"
          |     "pallet2" | "crate3" | "truck1"
```

**A:**
(lift-and-drive truck0 hoist0 crate0 pallet0 depot0 depot0) (lift hoist2 crate2 crate1 distributor1) (drive truck0 depot0 distributor0) (drive-and-lift truck0 hoist1 crate1 pallet2 distributor0) (drop hoist1 crate1 crate3 distributor0) (drive-and-load truck1 hoist0 crate0 depot0) (drive-and-unload truck1 hoist0 crate0 pallet2 depot0) (drive truck1 depot0 distributor1) (drive-and-load truck1 hoist2 crate2 distributor1) (drive-and-unload truck1 hoist2 crate2 pallet0 distributor1)

---

**Figure 11:** Prompt with real examples in the Depot domain from Pyperplan. The prompt template follows [69].

