# OpenReview forum: "Grammar Prompting for Domain-Specific Language Generation with  Large Language Models"
_NeurIPS.cc/2023/Conference — NeurIPS 2023 poster_

### Official Review · Reviewer_8y4i · 2023-06-25

**Soundness:** 3 good
**Presentation:** 4 excellent
**Contribution:** 2 fair
**Rating:** 5
**Confidence:** 3

**Summary:**

This paper studies generating strings from highly structured languages for large language models (LLMs). The authors propose grammar prompting to enhance LLMs to use external knowledge and domain-specific constraints, expressed through a grammar expressed in Backus–Naur Form. Serving as intermediate reasoning steps, these grammars (metalanguage) enhance the model's performance in generating highly structured languages in domains such as semantic parsing and molecule generation. The idea is simple and effective, and the experimental results well support the claim.

**Strengths:**

+ The paper is well-written and easy to follow.

+ A simple and effective method was presented, leading to improved downstream performance compared with simple prompting.

**Weaknesses:**

- The idea is not that novel, which uses metalanguage as the bridge that leads to the ultimate structured language. The method is an intuitive extension of the standard prompting (or chain-of-thought prompting) method.

- Improvements over simple prompting methods are validated, but whether the method can beat more sophisticated algorithms (e.g., the recent tree-of-thought reasoning [3]) is unknown. However, as the authors claimed, the focus of this paper is not to achieve SOTA in downstream tasks.

- The proposed constrained generation method is applied at the sub-sentence level, instead of the token level. Besides, the idea of constrained decoding has been explored in previous works [e.g., 1,2], which in turn challenges the novelty of this paper.


[1] Hokamp, Chris, and Qun Liu. "Lexically constrained decoding for sequence generation using grid beam search." arXiv preprint arXiv:1704.07138 (2017).
[2] Huang, Wenlong, et al. "Grounded decoding: Guiding text generation with grounded models for robot control." arXiv preprint arXiv:2303.00855 (2023).
[3] Yao, Shunyu, et al. "Tree of thoughts: Deliberate problem solving with large language models." arXiv preprint arXiv:2305.10601 (2023).

**Questions:**

NA

---

> ### Author Rebuttal · Authors · 2023-08-10
>
> We thank the reviewer for the feedback and helpful comments.
>
> We would like to emphasize the contribution and novelty of our work: (1) We focus on enabling LLMs to generate highly structured languages from just a few exemplars, a task where neither the standard nor the chain-of-thought prompting proves sufficient.  (2) Our method, grammar prompting along with the constrained decoding algorithm, is a novel and practical method that efficiently solves the grammar and program generation problem. (3) Our experiments on tool use, molecule generation, and PDDL planning are the first extensive set of empirical studies on the general capabilities of LLMs for data-efficient generation across a broad range of structured languages. We will make the novelty and contributions clearer in the new version of this paper.
>
> > The idea is not that novel, which uses metalanguage as the bridge that leads to the ultimate structured language. The method is an intuitive extension of the standard prompting (or chain-of-thought prompting) method.
>
> While our method can be viewed as “an intuitive extension of the standard prompting”, our motivation, problem (as mentioned above), and ideas are very different from those of standard prompting. To solve the grammar and program generation problem, we drew inspiration from programming language design, where BNF has been a standard protocol, but its use as an intermediary form for DSL generation was never unexplored in the machine learning and natural language processing literature (to the best of our knowledge). Moreover, by generating specialized grammar and programs constrained to the grammar, we can effectively detect the compliance of the program to the grammar, and potentially provide a diagnosis on the validity of the generated program. While chain-of-thought prompting generates the intermediate “thought”, the validity of the answer cannot be directly verified from the “thought” process.
>
> > Improvements over simple prompting methods are validated, but whether the method can beat more sophisticated algorithms (e.g., the recent tree-of-thought reasoning [3]) is unknown.
>
> The problem that our grammar prompting method targets are those with a combinatorial output space characterized by domain-specific grammars. Although the "tree-of-thought" approach is gestured toward broad problem-solving capabilities, instantiating it for more complex search challenges is nontrivial. For instance, grammar prompting employs the Earley algorithm for efficient traversal and potential backtracking within the search space. In comparison, naive usage of BFS or DFS in the tree-of-thought would require a significantly higher number of LLM calls for exploration and backtracking. And please kindly note that the preprint of the tree-of-thought reasoning paper was released after the NeurIPS deadline.
> > The proposed constrained generation method is applied at the sub-sentence level, instead of the token level. Besides, the idea of constrained decoding has been explored in previous works [e.g., 1,2], which in turn challenges the novelty of this paper.
>
> Generating specialized grammar and constrained programs at the sub-sentence level is a major strength of our work, instead of a limitation. Prior studies typically assume unrestricted access to smaller models (e.g., NMT models in 1), whereas our approach focuses on scenarios with limited and costly access to much larger blackbox models like GPT-3.5 and Codex. Constraining at the sub-sentence level can significantly reduce the number of LLM calls, thereby reducing the associated costs of employing LLMs.
>
> The constraints discussed in [2] are much simpler than the DSL constraints we specify using context-free grammars. And unfortunately, their constrained decoding cannot generalize to problems examined in this paper.
>
> Thank you again for bringing to our attention that we can make our contributions and novelty clearer. We will make sure to include the above discussions in the paper. We also really appreciate the pointers to the related work and we will cite in the paper.

---

### Official Review · Reviewer_pwxh · 2023-06-28

**Soundness:** 4 excellent
**Presentation:** 3 good
**Contribution:** 4 excellent
**Rating:** 8
**Confidence:** 5

**Summary:**

The authors investigate the effectiveness of grammar prompting as a simple approach to help LLMs utilize external knowledge and domain-specific constraints. This approach is motivated by the goal of enabling LLMs to generate DSL outputs that differ significantly from those encountered during pretraining. The authors achieve this by using a BNF grammar during in-context learning. In their framework, the LLM first predicts a BNF grammar based on a test input and then generates output constrained by the rules of the grammar. The experiments conducted demonstrate that grammar prompting enables LLMs to perform competitively on a diverse range of DSL generation tasks, such as semantic parsing, PDDL planning, and molecule generation.


**Strengths:**

- Novel method for prompting and constraining LLM generation using BNF grammar.
- Strong experimental results.
- The method is described clearly and is sound.
- The method is relatively simple yet effective. The framework is refreshing and is expected to have a significant impact on the semantic parsing community.


**Weaknesses:**

Minor: The contributions and novelty should be stated at the end of the introduction section.


**Questions:**

Have you explored other grammar forms?


**Limitations:**

No negative societal impacts

---

> ### Author Rebuttal · Authors · 2023-08-10
>
> We thank the reviewer for the positive feedback and constructive suggestions.
>
> > Minor: The contributions and novelty should be stated at the end of the introduction section.
>
> We will make the contributions/novelty clearer in the introduction.
>
> > Have you explored other grammar forms?
>
> We explored only the BNF meta-syntax formalism and did not consider alternatives (e.g., Wirth syntax notation), which are less commonly used in practice to describe syntaxes. However, this is a very interesting avenue and we hope to explore this direction in our future work.

---

> > ### Comment · Reviewer_pwxh · 2023-08-18
> >
> > Thanks for the response. I will keep the current ratings based on the responses and other reviews

---

### Official Review · Reviewer_rfkJ · 2023-07-05

**Soundness:** 4 excellent
**Presentation:** 4 excellent
**Contribution:** 3 good
**Rating:** 7
**Confidence:** 4

**Summary:**

This paper presents an approach for improving few-shot prompting of LLMs for tasks that produce structured outputs, where the structure can be described by a grammar in BNF form. The approach is similar to chain-of-thought prompting: for a given input x, the LLM is prompted to generate the minimal subset of grammar rules that can produce the corresponding output G[y], then conditions on x and G[y] to produce y. Decoding can be done in the standard way (e.g. greedy decoding), but the paper also presents an improved version that uses a modified Earley parsing algorithm to ensure that the outputs y conform to the predicted grammar G[y]. All variants of the approach are effective, providing substantial benefits over reasonable baselines (e.g. standard few-shot prompting and constrained decoding) across a range of tasks, including realistic semantic parsing benchmarks, SMILES molecule generation, and plan initialization for PDDL planning.

**Strengths:**

The approach is elegant and seems simple to implement, treating the LLM as a black box and needing only knowledge of the output grammar. While the constrained generation procedure is a bit more complex (and requires multiple LLM calls), it's also well-motivated and elegant, and isn't required for strong performance. I could definitely see the overall approach being practically useful in tasks that require DSLs that were low-resource in pre-training, such as tool use.

The experiments are thorough, with strong results across a range of tasks and datasets. The compositional generalization results on GeoQuery were especially interesting to me, as they indicate that the grammar examples in the prompt don't need to fully cover the grammar rules used in the output.

The paper is very clearly written, especially given the number of experiments presented and space constraints (but see below for a few questions).

**Weaknesses:**

One minor weakness is that it would benefit the paper to also show results on other LLMs to give evidence that the approach is broadly applicable, especially since of the three models evaluated, GPT-3.5 derives from Codex (I believe, if the paper is using code-davinci-002) and GPT-4's training data is unknown. StarCoder-15B could be one such model, although it may not be few-shot promptable.

A few details could be made clearer, if space is available, see below.

**Questions:**

Q1) How are the examples being retrieved in the retrieval-based ICL experiments? Do they use similarity of the generated grammar as well, or just the inputs?

Q2) Which Codex model is being used in the experiments here?

Minor clarification questions/points (don't need to address in author response):
- Be more consistent with the bolding of numbers in results tables, e.g. in Table 5
- The limitations mentions a lack of improvement for regexes, were these in experiments not reported in this paper?
- I didn't fully understand the Macro experiments on the PDDL tasks (although I did not check the appendix carefully).
- spelling of "conditioned" in Fig 2 caption
- while the semantic parsing experiments are well-motivated by tool use, they aren't really tool use IMO, so renaming 4.1 might be appropriate


**Limitations:**

The paper's limitation section did a good job, I think.

One additional potential limitation, though, is that, like in chain-of-thought prompting, the approach effectively introduces a pipeline: the model needs to first predict the grammar rules for an example, then condition on these rules to predict the output. This could lead to error propagation if the first step is wrong. However, the results are strong enough to indicate this probably isn't happening, and if it did it could be addressed with self-consistency / consensus decoding or another MBR-like approach.

---

> ### Author Rebuttal · Authors · 2023-08-10
>
> We thank the reviewer for the positive feedback and the recognition of the strengths of our work.
>
>
> > “One minor weakness is that it would benefit the paper to also show results on other LLMs …”
>
> In our recent experiments with PaLM2-Large using grammar prompting, we utilized the same 100 examples mentioned in Table 3. Our results indicate that grammar prompting outperforms the standard prompting baseline in two of the three benchmarks, though the performance improvement is less than was the case with the GPT family of models.
>
>
> | | GeoQuery | SMCalFlow | Overflow-Blk |
> |:------:|:-----:| :-----:|:-----:|
> | Standard Prompting | 90 | 14 | 59 |
> | Grammar Prompting | 87 | 17 | 62|
>
>
> > Q1) How are the examples being retrieved in the retrieval-based ICL experiments? Do they use similarity of the generated grammar as well, or just the inputs?
>
> Following prior work (e.g., Qiu et al. ’22), examples are retrieved based on their BM25 scores, using only the input natural language (NL) questions to calculate these scores. Although employing generated grammar for retrieval might enhance the quality of the examples obtained, it necessitates a minimum of two calls of an LLM per instance. In comparison, relying on NL questions for retrieval usually demands just a single call of LLM.
>
> > Q2) Which Codex model is being used in the experiments here?
>
> We used code-davinci-002 in the experiments.
>
> > Minor clarification questions
>
> Thank you for pointing out, and we will clarify these in our revised version.

---

> > ### Comment · Reviewer_rfkJ · 2023-08-12
> >
> > Thanks to the authors for the response! I still feel very positively about this paper after reading it and the other reviews and responses.

---

### Official Review · Reviewer_qhZ7 · 2023-07-10

**Soundness:** 3 good
**Presentation:** 3 good
**Contribution:** 3 good
**Rating:** 5
**Confidence:** 3

**Summary:**

This work explores grammar prompting as a simple approach for enabling LLMs to use external knowledge and domain-specific constraints, expressed through a grammar expressed in Backus–Naur Form (BNF), during in-context learning. Grammar prompting augments each demonstration example with a specialized grammar that is minimally sufficient for generating the particular output example, where the specialized grammar is a subset of the full DSL grammar.

The authors apply grammar prompting to various domain specific languages for semantic parsing (SMCalFlow, Overnight, GeoQuery), AI planning (PDDL), and molecule generation (SMILES), and find that it can meaningfully improve upon standard prompting baselines in the few-shot setting.

**Strengths:**


This paper is interesting and well-written.

This work is an efficient prompting generation algorithm for LLM.

Experimental results show that the proposed method seems effective in generating sequences with grammars.

**Weaknesses:**




In experiment part, how do you ensure the results are enough to verify the proposal? For example, the superiority in molecule generation is not well established compared with the other graph-based generation methods.



**Questions:**

See above.

**Limitations:**

See above.

---

> ### Author Rebuttal · Authors · 2023-08-10
>
> We thank the reviewer for the positive feedback and the recognition of the strengths of our work.
>
> > “In experiment part, how do you ensure the results are enough to verify the proposal? For example, the superiority in molecule generation is not well established compared with the other graph-based generation methods.”
>
> In molecule experiments, we focus on low-data class-specific molecule generation, as opposed to general molecule generation with massive amounts of data. In the low-data regime, recent work [29] has verified that current graph-based models fall short significantly compared with their grammar-based model, in the same settings as our molecule experiments. Our grammar prompting further improves upon [29] and thereby outperforms graph-based models in the specific low-data regime.
>
> Furthermore, our research reveals the initial success of applying LLMs in molecule generation tasks, suggesting a potentially fruitful avenue for future exploration.

---

### Author Rebuttal · Authors · 2023-08-09

We appreciate all reviewers' time and efforts in reviewing our paper. We are glad to find that reviewers generally recognized our key contributions and clear presentation of our paper:

**Method**: “The approach is elegant and seems simple to implement” [rfkj], “Novel method for prompting and constraining LLM generation using BNF grammar.” [pwxh], “This work is an efficient prompting generation algorithm for LLM.” [qhz7], “A simple and effective method was presented,” [8y4i]

**Experiment**: “The experiments are thorough, with strong results across a range of tasks and datasets.”[rfkj], “Strong experimental results.” [pwxh]

**Presentation**:  “The paper is very clearly written” [rfkj], “The method is described clearly” [pwxh], “The paper is well-written and easy to follow.” [8y4i], “This paper is interesting and well-written.” [qhz7]

Also, we thank all reviewers for their valuable and constructive suggestions. We reply to each reviewer individually below.

---

### Decision · Program_Chairs · 2023-09-21

**Decision:**

Accept (poster)

**Comment:**

The paper considers few-shot prompting. It demonstrates that using a specialized grammar (under which the correct output can be generated) as an intermediate step instead of natural language thoughts can yield an improvement on domain-specific text generation such as semantic parsing, planning, and molecule generation. The paper develops a novel scheme for constrained decoding based on an Earley parser. The improvement is very clear. The reviews are mainly positive and the authors' response is adequate. The paper, as in the case with many papers on prompting, does come with a certain risk of being model-specific (i.e., results are tied to specific LLMs) and in this case domain-specific, thus arguably narrow in scope. But under the risk, the results are sufficiently positive to justify an acceptance.